# A Nine-Gene Expression Signature Distinguished a Patient with Chronic Lymphocytic Leukemia Who Underwent Prolonged Periodic Fasting

**DOI:** 10.3390/medicina59081405

**Published:** 2023-07-31

**Authors:** Luca Emanuele Bossi, Cassandra Palumbo, Alessandra Trojani, Agostina Melluso, Barbara Di Camillo, Alessandro Beghini, Luca Maria Sarnataro, Roberto Cairoli

**Affiliations:** 1Department of Hematology and Oncology ASST Grande Ospedale Metropolitano Niguarda, 20162 Milan, Italy; alessandra.trojani@ospedaleniguarda.it (A.T.); agostina.melluso@ospedaleniguarda.it (A.M.); lucamaria.sarnataro@gmail.com (L.M.S.); roberto.cairoli@ospedaleniguarda.it (R.C.); 2Department of Information Engineering, University of Padova, 35020 Padua, Italy; barbara.dicamillo@unipd.it; 3Department of Comparative Biomedicine and Food Science, University of Padova, 35020 Padua, Italy; 4Department of Health Sciences, University of Milan, 20146 Milan, Italy; alessandro.beghini@unimi.it

**Keywords:** Chronic Lymphocytic Leukemia, gene expression signature, prolonged periodic fasting, microarrays, clustering, minimum spanning tree, lymphocytosis, B cells

## Abstract

*Background and Objectives*: This study aimed to investigate the causes of continuous deep fluctuations in the absolute lymphocyte count (ALC) in an untreated patient with Chronic Lymphocytic Leukemia (CLL), who has had a favorable prognosis since the time of diagnosis. Up until now, the patient has voluntarily chosen to adopt a predominantly vegetarian and fruitarian diet, along with prolonged periods of total fasting (ranging from 4 to 39 days) each year. *Materials and Methods*: For this purpose, we decided to analyze the whole transcriptome profiling of peripheral blood (PB) CD19+ cells from the patient (#1) at different time-points vs. the same cells of five other untreated CLL patients who followed a varied diet. Consequently, the CLL patients were categorized as follows: the 1st group comprised patient #1 at 20 different time-points (16 time-points during nutrition and 4 time-points during fasting), whereas the 2nd group included only one time point for each of the patients (#2, #3, #4, #5, and #6) as they followed a varied diet. We performed microarray experiments using a powerful tool, the Affymetrix Human Clariom™ D Pico Assay, to generate high-fidelity biomarker signatures. Statistical analysis was employed to identify differentially expressed genes and to perform sample clustering. *Results*: The lymphocytosis trend in patient #1 showed recurring fluctuations since the time of diagnosis. Interestingly, we observed that approximately 4–6 weeks after the conclusion of fasting periods, the absolute lymphocyte count was reduced by about half. The gene expression profiling analysis revealed that nine genes were statistically differently expressed between the 1st group and the 2nd group. Specifically, *IGLC3*, *RPS26*, *CHPT1*, and *PCDH9* were under expressed in the 1st group compared to the 2nd group of CLL patients. Conversely, *IGHV3-43*, *IGKV3D-20*, *PLEKHA1*, *CYBB*, and *GABRB2* were over-expressed in the 1st group when compared to the 2nd group of CLL patients. Furthermore, clustering analysis validated that all the samples from patient #1 clustered together, showing clear separation from the samples of the other CLL patients. *Conclusions*: This study unveiled a small gene expression signature consisting of nine genes that distinguished an untreated CLL patient who followed prolonged periods of total fasting, maintaining a gradual growth trend of lymphocytosis, compared to five untreated CLL patients with a varied diet. Future investigations focusing on patient #1 could potentially shed light on the role of prolonged periodic fasting and the implication of this specific gene signature in sustaining the lymphocytosis trend and the favorable course of the disease.

## 1. Introduction

Chronic Lymphocytic Leukemia (CLL) is a monoclonal disorder characterized by a progressive accumulation of functionally incompetent lymphocytes in blood, bone marrow, lymph nodes, and spleen. According to the Surveillance, Epidemiology, and End Results (SEER) database, the reported incidence rate of CLL is 4.9 cases per 100,000 individuals per year, making it one of the most prevalent forms of leukemia in the US and Europe. However, CLL is exceptionally uncommon in Asian populations [1]. The median age at the time of diagnosis is 70 years [2]. Approximately 10% of CLL patients are under the age of 45 [1]. In Europe, data indicates an annual incidence rate of 5.87 and 4.01 cases per 100,000 population among men and women, respectively [3]. CLL is sometimes incidentally discovered during routine blood cell count tests conducted for other reasons. The most common hematological characteristic of CLL is the absolute lymphocyte count (ALC) in the peripheral blood (PB) with median values ranging from 35 to 50 × 10^9^/L, accompanied by a level of ≥5000 B-lymphocytes/μL sustained for a minimum of 3 months. [4]. The Rai and Binet systems are commonly utilized to classify the clinical staging of CLL, relying on parameters such as spleen and liver size, platelet count, hemoglobin values, and the number of involved lymph nodes [5,6,7]. Peripheral blood flow cytometry is the most frequently employed test to confirm the diagnosis, aiming to detect the presence of circulating clonal B-lymphocytes expressing CD5^+^, CD19^+^, CD20^+^, and CD23^+^ [1,4,8]. Subsequent physical examination or computed tomography scan may reveal enlarged lymph nodes, splenomegaly, and hepatomegaly. As the disease progresses, common symptoms may manifest, including weight loss, fever, drenching night sweats, fatigue, abdominal fullness with early satiety, and increased susceptibility to infections [4]. Based on the variable region gene of the heavy immunoglobulin chain (*IGHV*) genes, patients can be clinically divided into two main subgroups: those with unmutated *IGHV* (UM-IGHV) exhibiting an aggressive form of the disease, and those with mutated *IGHV* (MT-IGHV) with a more favorable prognosis [8]. Approximately 80% of all patients exhibit one of the following four most common chromosomal abnormalities: del(13q14.3) (present in 55% of patients with a favourable prognosis if not associated with any other aberrations), trisomy 12 (found in 10–20% of cases; when occurring alone, it is associated with intermediate risk), del(11q23) (detected in 18% of patients with an unfavourable prognosis), and del(17p13) (associated with high risk in <10% of patients at the time of diagnosis and up to 30% in relapse/refractory cases; linked to *TP53* loss) [1,3]. Deletions and/or mutations of *TP53* have both prognostic and therapeutic implications as they can influence the disease’s progression and affect the decision-making process regarding patient treatment [8,9]. 

Several previous studies have identified biological pathways associated with common mutations in CLL. These mutations involve DNA-damage and cell-cycle-control (*TP53, ATM,* and *POT1*), chromatin modification (*HIST1H1E*, *HIST1H1B*, *CHD2*, *ZMYM3*, *BAZ2A*, *ASXL1*, *SYNE1*, *ARID1*, *KMT2D*, and *SETD2*), and mRNA and ribosomal processing (*SF3B1*, *XPO1*, *RPS15*, *DOX3X*, *ZNF292*, *MED12*, *CNOT3*, *U1*, *FUBP1*, *DDX3X*, *ZNF292*, and *NXF1*) [3,10]. Furthermore, recent studies have highlighted the role of the following key pathways in CLL: WNT signalling and MYC-related (*MGA* and *PTPN11*), Notch signalling (*NOTCH1* and *FBXW7*), and the inflammatory cascade (*MYD88*, *NFKBIE, BIRC3, TRAF3,* and *SAMHD1*) [3,10].

Additionally, mutations in BCR signalling (*EGR2*, *PAX5*, *BCOR*, *IRF4,* and *IKZF3)* and the MAPK–ERK pathway (*PTPN11*, *BRAF*, *KRAS, MAP2K1*, and *NRAS)* can influence cell-related B signalling and transcription mechanisms [2,3,10,11].

Gene expression profiling (GEP) studies have elucidated the pathophysiology of cells and offered valuable insights into the biomolecular mechanisms governing leukemia cell growth and survival, as well as predicting early progression in patients with CLL [12,13]. In our previous GEP study involving CLL patients (*n* = 112) we conducted the analysis based on the *IGHV* mutational status and *ZAP-70* (Zeta-chain-associated protein kinase 70) expression.

The comparison between two cohorts of CLL patients, namely *MTIGHV/ZAP70*^−^ and *UMIGHV/ZAP70^+^*, revealed significant dysregulation in the expression of genes encoding enzymes that regulated lipid metabolism, including *CHPT1*, *ARSD*, *LPL*, *AGPAT2*, *MBOAT1*, *AGPAT4*, *PLD1*, and *APP* [14]. Some authors have corroborated the crucial role of certain genes previously identified in our studies, such as *LPL* (lipoprotein lipase) and *ZAP-70* (Zeta-chain-associated protein kinase 70) [15,16]. Indeed, *LPL* expression exhibited a strong correlation with *IGHV* mutational status and overall survival, underscoring the significance of *LPL* as a prognostic marker in CLL [15]. Our study commenced by observing an untreated patient (#1) with CLL, characterized by a favorable prognosis, who voluntarily adopted a predominantly vegetarian and fruitarian diet, along with prolonged periods of total fasting (ranging from 4 to 39 days) since the time of diagnosis.

Patient #1 received a CLL diagnosis in 2016, and since then, continuous deep fluctuations in the ALC have been observed. Consequently, we sought to delve into this phenomenon and conducted an investigation into the transcriptomic profiling of B cells from this patient in comparison to the same cells of five other CLL patients (#2, #3, #4, #5, and #6) who followed a varied diet. The deliberate selection of five untreated CLL patients was aimed at conducting a comparison with patient #1, thereby avoiding any pharmacological interferences that could impact gene expression profiling evaluations. The aim of this study was to determine whether the nutrition, combined with prolonged periodic fasting adopted by patient #1, could potentially play a role in the observed trends of lymphocytic fluctuations and the benign course of the disease through changes in gene expression identified in the transcriptome profiling.

## 2. Materials and Methods

### 2.1. Patients 

We examined six patients with CLL (patient #1, #2, #3, #4, #5, and #6) who received their diagnosis at the Department of Hematology and Oncology (ASST Grande Ospedale Metropolitano Niguarda) based on standard morphological and immunophenotypical criteria. Written informed consent was obtained from all participants. Each of the six patients was of Caucasian ethnicity, and the clinical characteristics and molecular markers for each patient are summarised in Table 1. 

### 2.2. Patient #1 

In July 2016, patient #1, aged 50 years, underwent a routine blood chemistry examination, which revealed the following test results: ALC: 11.84 × 10^9^/L; WBC: 15.78 × 10^9^/L; platelets: 22.1 × 10^9^/L, and hemoglobin: 13.9 g/dL. To confirm the diagnosis of CLL, the patient was referred by the attending physician in the Department of Hematology and Oncology (ASST Grande Ospedale Metropolitano Niguarda). Prior to the diagnosis of CLL, the patient had a history of good health, with the exception of an appendectomy due to peritonitis at the age of 9. There was no known family history of CLL, although his father had been diagnosed with non-Hodgkin’s lymphoma. Currently, the patient does not take any drugs, does not frequently consume alcoholic beverages, and is an ex-smoker with regular lifestyle habits.

#### 2.2.1. Diagnostic Assessment

The morphological examination of PB cells revealed the presence of Gumprecht shadow cells (smudge cells). Flow cytometry confirmed the presence of monoclonal K (low- intensity) B cell lymphocytosis and a phenotype of CD5+, CD19+, CD23+. FISH analysis detected an interstitial deletion (13q), mutated *IGHV*, and wild type *TP53*. The patient’s CLL was classified as stable Rai stage 0 and Binet stage A. Since the time of diagnosis until the present, the patient has remained asymptomatic, and there has been a slow-growth trend of lymphocytosis, but treatment has not been necessary.

#### 2.2.2. Nutrition and Fasting Periods

Since 2016, in accordance with his attending physician, he has been undergoing annual follow-up, and blood chemistry assessments every four months, following a “watch and wait” approach [4]. Since the time of diagnosis, he has voluntarily opted for a predominantly fruit and raw vegetable-based diet, periodically engaging in prolonged total fasting (ranging from a minimum of 4 days to a maximum of 39 days) during which he consumes only water and herbal tea. The most recent fasting period took place from 26 December 2022, to 4 January 2023 (9 days) (Figure 1).

#### 2.2.3. Lymphocytosis 

Since 2016, we have been monitoring the blood tests of patient #1 to assess the absolute lymphocytes values and to observe the fluctuations and the trend of lymphocytosis. 

### 2.3. Patients #2, #3, #4, #5, and #6

Patients #2, #3, #4, #5, and #6 were diagnosed in 2019, 2018, 2018, 2009, and 2009, respectively. All of them underwent a maximum of 3 evaluations per year, following the standard good clinical practice for Chronic Lymphocytic Leukemia (Table 2). We selected the B cells from each patient simultaneously in 2021.

### 2.4. Selection of B Cells

We collected Peripheral Blood Mononuclear Cells (PBMCs) from all six patients with CLL. The patients were divided into two groups as follows: in the 1st group, we selected CD19+ cells from blood samples of patient #1 at 20 time-points (16 time-points during nutrition and 4 time-points during fasting). As for the 2nd group, we opted for only 1 time-point for each patient (#2, #3, #4, #5, and #6) due to their various nutritional regimens. PBMCs from all CLL patients were isolated using Lymphoprep^TM^ density gradient centrifugation (STEMCELL Technologies, Vancouver, BC, Canada) at 800 rpm for 20 min. Subsequently, CD19+ cells were purified from the PBMCs of all six patients using MACS CD19 Microbeads (Miltenyi Biotec, Bologna, Italy), following the manufacturer’s instructions. The CD19+ cells were then resuspended in 50 μL of RNA *later* (Thermo Fisher, Milano, Italy) and stored in a cell bank at −20 °C until RNA extraction was performed [14].

### 2.5. Total RNA Preparation

Gene expression profiling was conducted on total RNA extracted from CD19+ cells of the six patients using MagMAX 96 Total RNA Isolation Kit (Thermo Fisher Scientific), according to the manufacturer’s instructions. The quality and yield of the extracted RNA were assessed using the NanoDrop™ 2000 Spectrophotometers (Thermo Fisher Scientific).

### 2.6. Gene Expression Profiling Experiments

We performed microarray analyses on BM CD19+ cell samples (*n* = 20) from patient #1 (as described earlier) and BM CD19+ cell samples (*n* = 5) from each of patients #2, #3, #4, #5, and #6. The whole transcriptome analysis of isolated RNA was carried out on the CD19+ cells of all CLL patients using the human Clariom™ D Pico Assay (Affymetrix, Thermo Fisher Scientific). We decided to prepare 25 arrays: 20 for the 1st group and 1 array for each patient in the 2nd group. For each sample, 5 ng/µL of total RNA in 20 µL of nuclease-free water was retrotranscribed into single-stranded cDNA containing a T7 promoter sequence at the 5′ end. Subsequently, a 3′ adaptor was added to the single-stranded cDNA, serving as a template for double-stranded cDNA synthesis in a pre-IVT amplification reaction. The reaction involved the use of DNA polymerase and RNase H to simultaneously degrade RNA and synthesize single-stranded cDNA with the 3′ adaptor. Taq DNA polymerase and specific primer adapters were then added to the single-stranded cDNA to synthesize and pre-amplify the double-stranded. The pre-IVT amplification reaction was optimized with 9 cycles of amplification, as previously reported by Taq DNA polymerase [18]. To synthesize cRNA, the double-stranded cDNA was used as a template, and the amplification reaction was carried out for 16 h at 40 °C, followed by cooling to 4 °C. The RT-IVT method was performed following the technique developed by Van Gelder et al. [19]. Subsequently, 20 µg of purified cRNA was resuspended in 24 µL nuclease-free water to obtain sense-strand cDNA, followed by RNase H digestion for single-stranded cDNA synthesis. After hydrolysis, 5.5 µg of single-stranded cDNA was resuspended in 46 µL of nuclease-free water in preparation for the next step of fragmentation and labeling. Fragmentation of single-stranded cDNA was achieved using uracil-DNA glycosylase (UDG) and apurinic/apyrimidinic endonuclease 1 (APE1); the labeling of single-stranded cDNA was performed by terminal Deoxynuocleotydil transferase (TdT). Next, 200 µL of the hybridization cocktail was injected into a single Whole Transcript 49-Format array and loaded into the Affymetrix GeneChip™ Hybridization Oven 645 with rotation at 60 rpm for 16 h at 45 °C. Finally, the arrays were washed and stained using an Affymetrix GeneChip™ Fluidics Station 450, following the fluidics protocol (FS450_0001), and scanned with an Affymetrix GeneChip™ Scanner. 

### 2.7. Bioinformatic Analysis of GEP Data

The Microarray CEL files were subjected to background correction, normalization, and summarization using the robust multiarray average (RMA) method [20]. Significance Analysis of Microarrays (SAM, version 3.0) was employed to assess significant differentially expressed genes between the 1st group and 2nd group of CLL patients [21]. Control for multiple testing was carried out using false discovery rate (FDR), employing the implementation proposed by Storey and Tibshirani [22]. Control probes and probes with an average expression level or variance below the first quartile were filtered out preliminary to enhance statistical power. Hierarchical agglomerative clustering was performed on the samples using Ward distance, considering only the differentially expressed genes. Finally, to further examine the similarities and differences among samples, we calculated the Euclidean pair-wise distance between samples based on the genes selected as differentially expressed. Using this distance as weight, we constructed a minimum spanning tree (MST), which is an undirected graph that connects all the nodes, i.e., the samples, together, without any cycles and with the minimum possible total edge weight (where an edge weight represents the Euclidean distance between samples). All analyses were conducted using R Studio (version 2022.07.2+576) software. 

## 3. Results

### 3.1. ALC and Lymphocytosis Trend of Patient #1

We monitored the absolute lymphocyte counts between 2016 and 2023, and we observed significant deep fluctuations of lymphocytosis during this period, as shown in Figure 2.

In 2021, we began to investigate this phenomenon further and observed that, between April and July 2021 (1–9) and between March and June 2022 (10–14), the ALC of patient #1 was approximately halved (Table 3). Particularly, during the period approximately 4 to 6 weeks post-fasting, the trend of lymphocytosis was on a decreasing trajectory, reaching minimal values.

### 3.2. ALC and Lymphocytosis of Patients #2, #3, #4, #5, and #6

As observed in Table 2, all patients in the 2nd group did not exhibit fluctuations in the absolute lymphocytes count but instead showed a growing trend of lymphocytes over time.

### 3.3. Cluster Dendrogram Patient #1 vs. Patients #2, #3, #4, #5, and #6 

The results of hierarchical clustering are depicted using a tree-based representation known as a dendrogram in Figure 3. Patients #2, #3, #4, #5, and #6 clustered together, independently from patient #1.

### 3.4. Nine Genes Were Differently Expressed in the CLL Patient Who Followed Prolonged Periodic Fasting vs. CLL Patients with a Varied Diet

Differential expression analysis, revealed, after correction for multiple testing, 18 differentially expressed genes (DEGs) in the comparison between CD19+ cells of patient #1 and patients #2, #3, #4, #5, and #6. Among them, we focused on nine genes with functional annotation: four genes were downregulated (*IGLC3*, *RPS26*, *CHPT1*, and *PCDH9*), and five genes were upregulated (*IGHV3-43*, *IGKV3D-20*, *PLEKHA1*, *CYBB*, and *GABRB2*) in the 1st group (patient #1) vs. the 2nd group of patients (#2, #3, #4, #5, and #6 (Table 4). In particular, *IGLC3*, *RPS26*, *PLEKHA1*, *CYBB*, and *GABRB2* exhibited a high fold change (FC) > 3. Among them, *RPS26*, *CHPT1*, *CYBB*, and *GABRB2* were found to be associated with KEGG pathways, as shown in Table 5. Additionally, we generated a heat map on log2 expression of the nine differentially expressed genes across different patients/conditions (Figure 4). False discovery rate (FDR) (q-value) and FC values are provided in Table 4.

### 3.5. Minimun Spanning Tree (MST) of Patient #1 vs. Patients #2, #3, #4, #5, and #6

As explained in the methods, we also computed the Euclidean pair-wise distance between samples of patient #1 (during nutrition and fasting) and samples of patient #2, #3, #4, #5, and #6 (following a varied diet), using this distance as the weight. Consequently, we constructed a minimum spanning tree (MST) that displays, within the same graph, the subjects as colored nodes and the pairs of closest subjects connected by edges. Black and red dots represent the samples from the 2nd group and the 1st group, respectively (Figure 5). Once again, patients #2, #3, #4, #5, and #6 are closer to each other than to any other sample belonging to patient #1.

## 4. Discussion

Here, we present a patient with CLL who self-referred to the Department of Hematology and Oncology at the ASST Grande Ospedale Metropolitano Niguarda since the time of diagnosis in 2016. Up until now, he has not required treatment and he voluntarily follows a mainly fruitarian and vegetarian diet, incorporating long periods of absolute fasting since 2016. Specifically, he observed periods of absolute fasting, consuming only water and herbal tea, as follows: 2017 (39 days), 2018 (17 days), 2019 (21 days), 2020 (27 days), 2021 (33, 4, 4, and 6 days), 2022 (26, 19, and 5 days), and in 2023 (4 days). We directed our attention to the lymphocytosis, which revealed repetitive fluctuations in the absolute lymphocyte counts since diagnosis (Figure 2). Interestingly, we observed that approximately 4 to 6 weeks after the end of fasting, the absolute lymphocyte count was reduced by about half, as shown in Table 3. 

Our study aimed to explore whether these significant deep fluctuations in the absolute lymphocyte count could be attributed to the periods of fasting. To achieve this goal, we opted to conduct a highly detailed examination of the whole transcriptome profiling using microarray analysis with the human Clariom™ D Pico Assay, one of the most recent and powerful tools for generating high-fidelity biomarker signatures. We conducted a whole transcriptome analysis on CD19+ cells obtained from patient #1 under different nutritional conditions (nutrition and fasting) vs. the same cells from five untreated CLL patients who followed a varied nutrition regimen without fasting at a single time point, as detailed in the materials and methods section. 

A potential limitation of our study is the small sample size, as well as the absence of trend analyses for the absolute lymphocyte count in both groups. These factors could restrict the establishment of a cause–effect relationship and the generalizability of our findings. However, we plan to conduct GEP experiments on a larger cohort of patients with CLL to evaluate the molecular signature that distinguishes group 1 from group 2. Currently, we are unable to investigate other CLL patients with the same unique characteristics as patient #1, but we will continue to monitor and collect B cell samples from him as he maintains prolonged fasting periods. 

Hierarchical clustering analysis revealed that patient #1, during nutrition and fasting, (1st group) clustered separately from all CLL patients (2nd group), indicating a gene expression signature that differentiated patient #1 from patients #2, #3, #4, #5, and #6 as depicted in Figure 3. GEP results showed that nine genes were significantly differently expressed between patient #1 (1st group) and patients #2, #3, #4, #5, and #6 (2nd group). Among these genes, four genes were downregulated (*IGLC3, RPS26, CHPT1,* and *PCDH9*) and five were upregulated *(IGHV3-43*, *IGKV3D-20*, *PLEKHA1*, *CYBB*, and *GABRB2*) in the 1st group compared to the 2nd group. Consequently, we explored the potential role of these genes in CLL and cancer. 

The ribosomal protein S26 (*RPS26*) functions as a checkpoint for T lymphocyte survival and homeostasis in a p53-dependent manner, as reported by Chen et al. [23], and Zhao et al. [24], who demonstrated its upregulation in acute lymphoblastic leukemia (ALL) through qRT-qPCR analysis. In our study, we observed an overexpression of *RPS26* in CLL patients belonging to the 2nd group compared to the 1st group. 

In our previous GEP study involving 112 CLL patients, we found that *UMZAP70^+^* subjects exhibited upregulation of genes encoding enzymes involved in lipid metabolism, including *CHPT1* [14]. Additionally, we established that the downregulation of *CHPT1* in CLL correlated with a favorable prognosis [14]. In this study, we confirmed our previous GEP findings by demonstrating that *CHPT1* expression was significantly under expressed in the 1st group compared to the 2nd group of CLL patients. On the other hand, some studies have shown that the downregulation of *CHPT1* is associated with an unfavorable prognosis in certain types of solid tumors, such as breast cancer [25] and lung cancer [26]. As for lipid metabolism in CLL, several authors have reported a high incidence of hypercholesterolemia in CLL patients, and the use of statins may have an impact on the clinical course of the disease [27]. Furthermore, a study utilizing an in vitro model of the pseudofollicles from which CLL cells originate, demonstrated that low-density lipoproteins (LDLs) increased plasma membrane cholesterol and promoted CLL cell proliferation. The findings of this study suggested a potential survival benefit from cholesterol-lowering statin drugs in CLL patients [28].

Some authors have indicated that the downregulation, loss of expression, and deletion of *PCDH9* were associated with the progression of prostate cancer [29], epithelial ovarian cancer [30], glioma [31,32], and gastric cancer [33]. In our study, we showed the downregulation of *PCDH9* in the 1st group vs. the 2nd group of CLL patients, and further investigations will shed light on the potential role of *PCDH9* in CLL. As for *CYBB*, it has been relatively less studied. Some authors found that *CYBB* was upregulated in Chronic Myeloid Leukemia (CML) [34], while it served as a prognostic and diagnostic marker in myeloproliferative neoplasms (MPN) [35]. In our GEP study, we observed the upregulation of *CYBB* in the 1st group compared to the 2nd group of CLL patients. *CYBB* is involved in KEGG pathways related to cell death, such as ferroptosis and necroptosis, as presented in Table 5. 

*GABRB2* plays a crucial role in the formation of the GABA_A_ receptor in the mammalian brain, and its expression can vary in solid tumors. It has been found to be upregulated in salivary gland cancer [36], adrenocortical carcinoma [37], and lymph node metastases of papillary thyroid carcinoma [38]. Conversely, *GABRB2* was downregulated in colorectal cancer [39], brain tumors [40,41], and kidney tumors [42]. To the best of our knowledge, this is the first study to highlight changes in *GABRB2* expression in CLL, demonstrating that this gene was overexpressed in the 1st compared to the 2nd group of CLL patients. Numerous research studies have shown that fasting can confer various benefits, including regulation of metabolism (lipid, glucose, protein, and neuroendocrine), reduction in inflammation, and, concerning cancer, the inhibition of signals promoting cellular growth and survival [43,44]. Furthermore, some authors have proposed a potential role for prolonged periodic fasting during conventional cancer treatment, aiming to enhance therapeutic efficacy and limit side effects in cancer patients [44].

## 5. Conclusions

To the best of our knowledge, this is the first transcriptome study analyzing a patient with CLL who has followed a nutrition primarily based on vegetables and fruits, combined with long-term periodic fasting since diagnosis, and exhibited significant deep fluctuations in lymphocytosis. We presented a novel gene expression signature that distinguished patient #1 from five CLL patients who followed a varied diet. Future investigations should be conducted to elucidate the potential role of fasting in maintaining the absolute lymphocyte counts in patient #1. Additionally, we will explore whether this small signature could be valuable for future studies related to CLL prognostication.

## Figures and Tables

**Figure 1 medicina-59-01405-f001:**
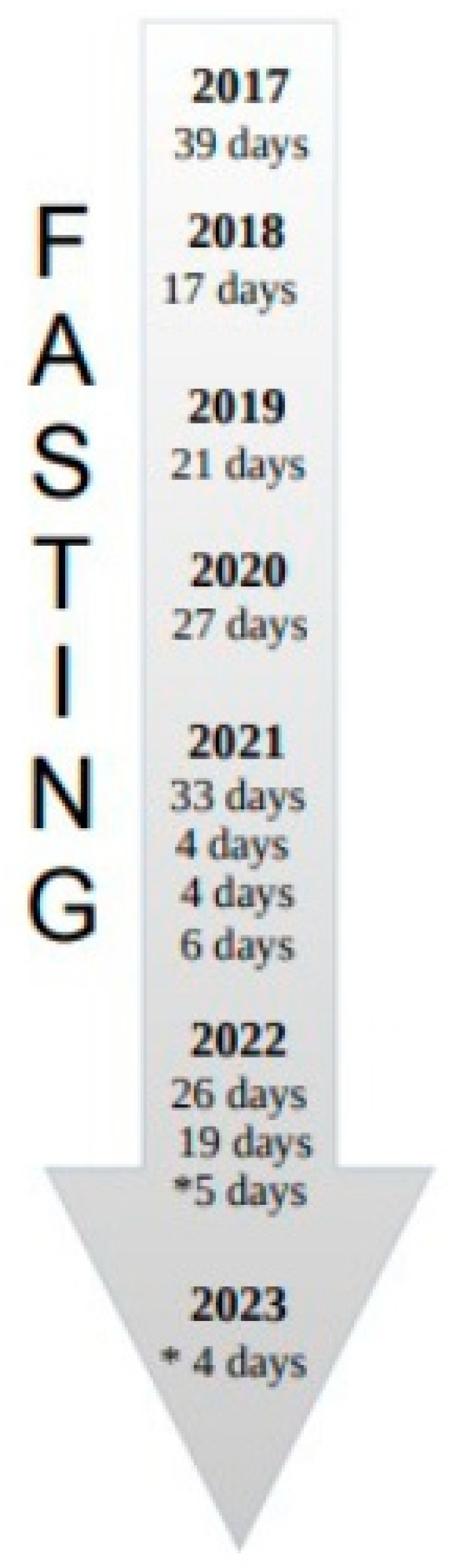
Fasting timeline of patient #1. He followed a strict fasting regime as follows: 39 days (2017), 17 days (2018), 21 days (2019), and 27 days (2020). In 2021, he fasted four times: 33, 4, 4, and 6 days, respectively. From 2022 to 2023, he fasted for a total of 9 days (*5 days in 2022 and *4 days in 2023, respectively).

**Figure 2 medicina-59-01405-f002:**
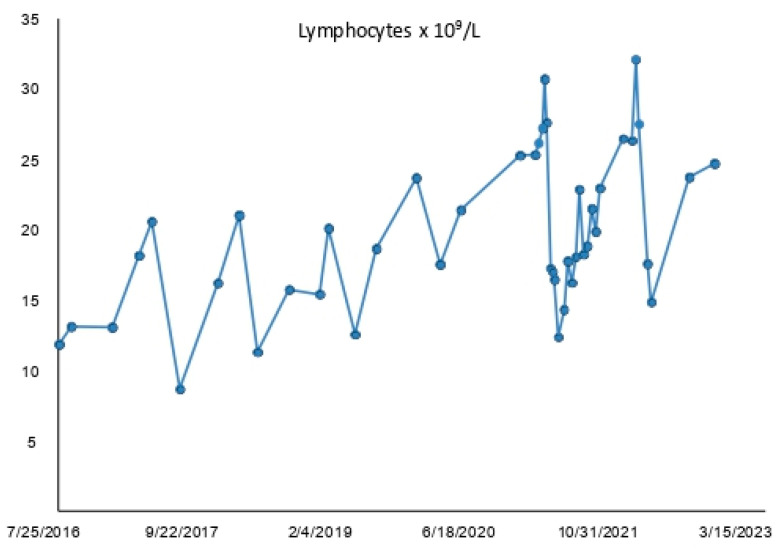
Fluctuations of absolute lymphocytosis in patient #1 since diagnosis. The lymphocyte levels exhibited varying trends between July 2016 and January 2023.

**Figure 3 medicina-59-01405-f003:**
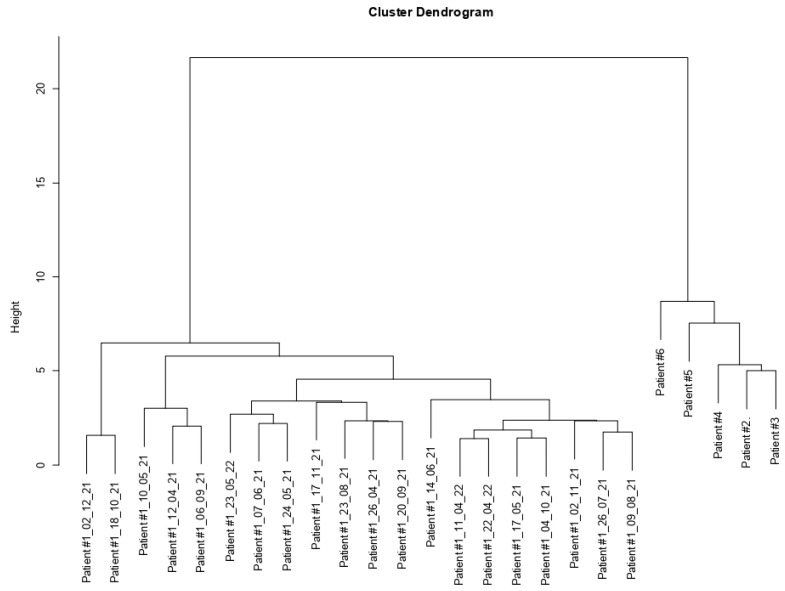
Clustering results. The figure illustrates the outcomes of clustering (hierarchical agglomerative clustering) achieved by grouping samples according to the similarities in gene expression of the differentially expressed genes.

**Figure 4 medicina-59-01405-f004:**
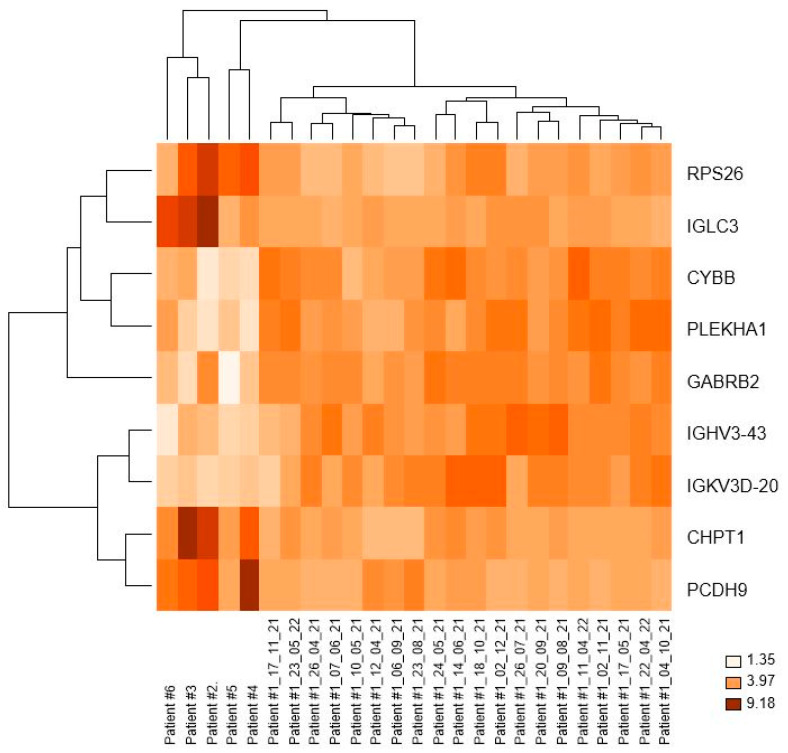
Heatplot generated based on the log2 expression of the nine differentially expressed genes across different patients/conditions.

**Figure 5 medicina-59-01405-f005:**
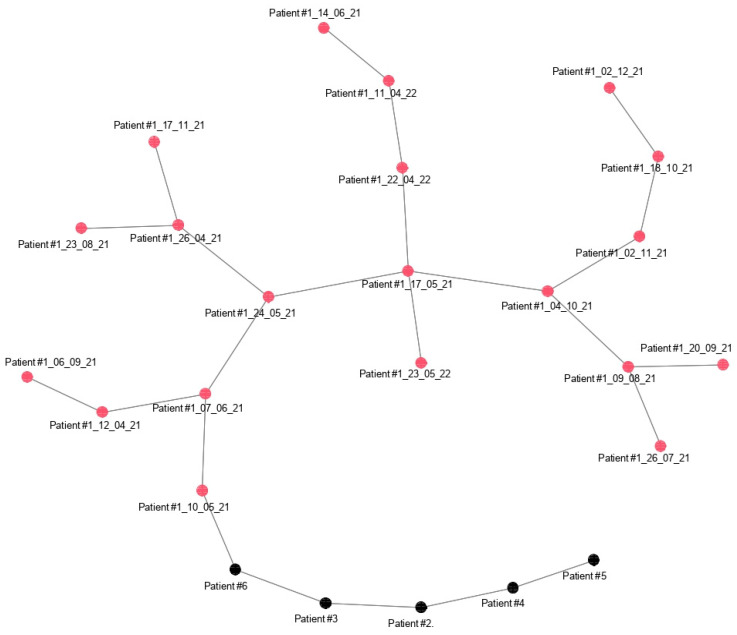
Minimum spanning tree that connects all the nodes, i.e., the samples together, without any cycles and with the minimum possible total edge weight (where an edge weight represents the Euclidean distance between samples).

**Table 1 medicina-59-01405-t001:** Clinical features and molecular markers of the six patients with CLL.

Patient	Age at Diagnosis (y)	Time from Diagnosis to B Cell Selection (y)	Sex	Binet Stage	Rai Stage	*IGVH* Gene	*TP53* Mutation Status	FISH *
#1	50	5	M	A	0	MT	UM	del(13q)
#2	64	2	M	C	IV	UM	UM	negative
#3	64	3	M	C	IV	UM	UM	trisomy 12; del(11q)
#4	70	3	M	B	II	UM	UM	trisomy 12
#5	51	12	M	B	II	MT	UM	negative
#6	70	12	F	B	II	MT	UM	monosomy 13q14

Abbreviations: M, male; F, female; MT, mutated; UM, unmutated. * According to the Döhner classification [17].

**Table 2 medicina-59-01405-t002:** The absolute values of the lymphocyte counts for patients #2, #3, #4, #5, and #6 from the time of diagnosis to the time of B cell selection.

Patient	Time-Points	Date	Lymphocyte Count × 10^9^/L
#2	1	June 2019 *	8.00
#2	2	December 2019	19.0
#2	3	May 2020	39.34
#3	1	February 2018 *	10.50
#3	2	October 2018	12.36
#3	3	April 2019	11.91
#3	4	October 2021	106.68
#4	1	November 2018 *	14.05
#4	2	December 2018	25.31
#4	3	June 2019	97.70
#4	4	August 2021	234.94
#4	5	October 2021	279.44
#5	1	May 2009 *	9.6
#5	2	August 2009	11.4
#5	3	November 2009	12.6
#5	4	June 2010	13.58
#5	5	February 2011	15.1
#5	6	October 2011	16.6
#5	7	May 2012	19.92
#5	8	January 2013	23.37
#5	9	May 2013	21
#5	10	December 2013	26.7
#5	11	May 2014	24.6
#5	12	June 2015	35.2
#5	13	November 2015	51.10
#5	14	December 2016	52.3
#5	15	June 2017	62.0
#5	16	June 2021	141.6
#6	1	May 2009 *	12.0
#6	2	August 2011	11.38
#6	3	January 2012	12.7
#6	4	June 2012	12.0
#6	5	January 2013	10.89
#6	6	June 2013	14.78
#6	7	December 2013	19.2
#6	8	June 2014	19.29
#6	9	December 2014	21.85
#6	10	June 2015	20.92
#6	11	December 2016	34.36
#6	12	January 2018	54.8
#6	13	June 2018	71.2
#6	14	February 2019	67.0
#6	15	September 2019	38.9
#6	16	April 2021	43.4
#6	17	October 2021	112.46

* Time of Diagnosis.

**Table 3 medicina-59-01405-t003:** The absolute values of the lymphocyte counts for patient #1 in relation to 2021 and 2022 during periods of fasting and nutrition.

Time-Points	Date	Nutrition and Fasting	Lymphocyte Count × 10^9^/L
1	12 April 2021	nutrition	25.32
2	26 April 2021	fasting	26.12
3	10 May 2021	fasting	27.21
4	17 May 2021	nutrition	30.71
5	24 May 2021	nutrition	27.59
6	7 June 2021	nutrition	17.24
7	14 June 2021	nutrition	17.01
8	21 June 2021	nutrition	16.44
9	5 July 2021	nutrition	12.35
10	28 March 2022	nutrition	26.3
11	11 April 2022	fasting	32.07
12	22 April 2022	fasting	27.47
13	23 May 2022	nutrition	17.55
14	6 June 2022	nutrition	14.82

**Table 4 medicina-59-01405-t004:** List of annotated differentially expressed genes in the 1st group (samples from patient #1) vs. 2nd group of patients (#2, #3, #4, #5, and #6). The table presents the gene symbol and name, the average expression value in logarithmic scale (base 2) for the 1st and 2nd group, the fold change (FC) in the 1st group with respect to the 2nd group, the *p*-value, and the q-value (*p*-value corrected based on the false discovery rate definition).

Gene Symbol	Gene Name	log-Mean Group 2	log-Mean Group 1	FC 1st Group vs. 2nd Group	*p* Value	q Value
*IGLC3*	immunoglobulin lambda constant 3 (Kern-Oz+ marker)	6.92	4.18	−6.67	0.0000136	0.04493
*RPS26*	ribosomal protein S26	6.71	4.83	−3.67	0.0000235	0.04926
*CHPT1*	choline phosphotransferase 1	3.72	2.29	−2.69	0.0000149	0.04396
*PCDH9*	protocadherin 9	2.88	1.73	−2.20	0.0000177	0.04219
*IGHV3-43*	immunoglobulin heavy variable 3-43	2.19	3.46	2.41	0.0000142	0.04362
*IGKV3D-20*	immunoglobulin kappa variable 3D-20	1.66	3.10	2.71	0.0000192	0.04460
*PLEKHA1*	pleckstrin homology domain containing, family A (phosphoinositide binding specific) member 1	2.72	4.36	3.13	0.0000173	0.04360
*CYBB*	cytochrome b-245, beta polypeptide	3.77	5.48	3.27	0.0000131	0.04505
*GABRB2*	gamma-aminobutyric acid (GABA) A receptor, beta 2	4.06	6.55	5.63	0.0000125	0.04876

**Table 5 medicina-59-01405-t005:** *RPS26*, *CHPT1*, *CYBB*, and *GABRB2* and KEGG pathways.

Gene Symbol	KEGG Pathway
*RPS26*	hsa03010 Ribosome
*CHPT1*	hsa00440 Phosphonate and phosphinate metabolism hsa00564 Glycerophospholipid metabolism hsa00565 Ether lipid metabolism hsa01100 Metabolic pathways hsa05231 Choline metabolism in cancer
*CYBB*	hsa04066 HIF-1 signaling pathway hsa04145 Phagosome hsa04216 Ferroptosis hsa04217 Necroptosis hsa04621 NOD-like receptor signaling pathway hsa04670 Leukocyte transendothelial migration
*GABRB2*	hsa04080 Neuroactive ligand–receptor interaction hsa04726 Serotonergic synapse hsa04727 GABAergic synapse

## Data Availability

All data involved in this work will be made available by the corresponding authors upon request (lucaemanuele.bossi@ospedaleniguarda.it or cassandra.palumbo@ospedaleniguarda.it).

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
