# Peer review of "A Nine-Gene Expression Signature Distinguished a Patient with Chronic Lymphocytic Leukemia Who Underwent Prolonged Periodic Fasting"

_medicina, 2023, doi:10.3390/medicina59081405_

Round 1

Reviewer 1 Report

Thank you for your interesting submission. Lipid metabolism has always been an interesting biological under pinning in CLL. 

Few questions about the samples.

1. When were the samples taken relative to diagnosis? this should be included in the table

2. Were there any racial differences between the patients? this should included or statement that it is not known included

3. When the fasting time points are removed is patient 1 more similar to the rest? this should be the first analysis.

4. Were the fasting periods the same during the years ( always the same months or seasonal variation? 

5. did the rise in ALC increase with vaccinations or viral illnesses? versus nutrition/ fasting? this should be stated as we see clear rise and falls of ALC associated with this.

Analysis Comments

6.from the previous data set, do the samples otherwise align and map to the cohort when the non fasting samples are removed?

7. were all the samples on a single array, or various arrays, how was the plate to plate variability accounted for, was there a sample patient sample accross all arrays?

8. A heat map of the gene expression with FDR and fold change should be presented as part of the main figures.

9. GSEA should be performed to look at the signature

10. A validation should be applied to the previous data sets and see if any patients cluster with them.

11. JAK/Stat pathways and steroid metabolism have been evaluated previously as well. The impact oF fasting should be reviewed in this context. EBioMedicine. 2017 Feb;15:24-35. Leuk Lymphoma. 2016;57(4):797-802.

none

Author Response

Dear Reviwer 1,

Thank you in advance

Kind Regards

Bossi Luca Emanuele

Reviewer 2 Report

THE THEME OF WORK COULD BE INTERESTING, HOWEVER I ASK THAT COMPARING A GROUP CONSISTING OF ONLY 1 PATIENT WITH ONE CONSISTING OF ONLY 6 PATIENTS DOES NOT ALLOW ANY CONCLUSION. THE DATA IS DEFINITELY DESERVING OF INTEREST BUT NEEDS SOME MORE NUMBERS IN A DISEASE AMONG THE MOST COMMON IN HEMAOTLOGY. THE AUTHORS COULD MAYBE LOOK FOR OTHER PATIENTS FOR GROUP 1 IN OTHER CENTERS. AT THE CURRENT STATE IN MY NOTICE IT CANNOT BE ACCEPTED.

Author Response

Dear Reviwer 2,

Thank you in advance

Kind Regards

Bossi Luca Emanuele

Round 2

Reviewer 2 Report

ok for the new version

Author Response

Dear Reviewer  1,

Thank you for your comments. We have attached the responses to the Editor's suggestions in the appropriate section.

Thank you in advance

Kind Regards

Bossi Luca Emanuele